

# Historical rainfall data in Northern Italy predict larger meteorological drought hazard than climate projections

Rui Guo and Alberto Montanari

Department of Civil, Chemical, Environmental and Materials Engineering (DICAM), University of Bologna, Bologna, Italy

**Correspondence:** Rui Guo (rui.guo2@unibo.it)

**Abstract.** Simulation of daily rainfall for the region of Bologna produced by 13 climate models for the period 1850 - 2100 are compared with the historical series of daily rainfall observed in Bologna for the period 1850 - 2014, and analysed to assess meteorological drought changes up to 2100. In particular, we focus on annual rainfall data, seasonality and drought events to derive information on the future development of critical events for water resources availability. The results prove
that rainfall statistics, including seasonal patterns, are fairly well simulated by models, while the historical sequence of annual rainfall is not satisfactorily reproduced. In terms of meteorological droughts, we conclude that historical data analysis under the assumption of stationarity may depict a more critical future with respect to climate model simulations, therefore outlining important technical indications.

## 1 Introduction

Droughts originate one of the most challenging risks for modern society. Indeed, most countries across the globe are exposed to medium/high drought risk (see Fig. 1). Recent events, like the Millenium Drought in Australia (Van Dijk et al., 2013) and droughts in California (Lund et al., 2018) and South Africa (Sousa et al., 2018) pointed out once again that drought events may occasionally last for several years, therefore originating "multiyear droughts", with 5-10 years duration, or "megadroughts", with duration longer than a decade (Cook et al., 2016). The sporadic occurrence of megadroughts across the globe is also
confirmed by several paleoclimatic records (Cook et al., 2016; Vance et al., 2015). Multiyear droughts and megadroughts are a reason of concern as they strongly impact ecosystems, water supply, socio-economical assets and, ultimately, public health (Ukkola et al., 2020; Tabari, 2020; Tabari et al., 2021; Stahl et al., 2016). Also, the vulnerability and exposure to these events are much higher than in the past.

   Multiyear droughts are rare extreme events, which are related to large scale atmospheric teleconnections whose dynamics
are dictated by chaotic behaviors. An implication of the rare occurrence of multiyear droughts is the limited availability of data to decipher their frequency and train prediction models. For the same reason, it is also difficult to predict how climate change may exacerbate drought risk. Indeed, the warmer climate is changing the hydrological cycle and further affects precipitation (Trenberth, 2011), with evidence demonstrating that precipitation is altering in terms of both annual mean (Knutti and Sedláček, 2013), seasonal variation (Polade et al., 2014; Kumar et al., 2014a), and extreme events (Papalexiou and Montanari, 2019;
Alexander et al., 2006). According to the Intergovernmental Panel on Climate Change Sixth Assessment Report (IPCC AR6),





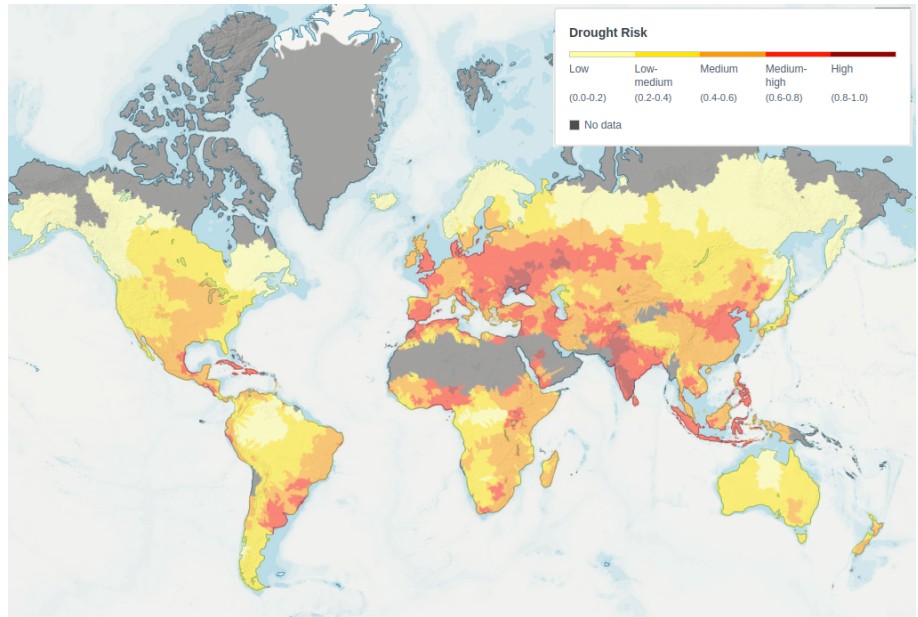

**Figure 1.** Drought risk index score by countries. It combines information on droughts hazard, exposure, and vulnerability. Higher values indicate higher risk of drought. Source: WRI Aqueduct, accessed on Aug 8, 2022 (aqueduct.wri.org).

such changing conditions are likely to continue in the 21st century (IPCC, 2021). Therefore, the risk of multiyear droughts is likely to increase in the future, in view also of the increased water demands originated by global warming (Li et al., 2020).

The above reasoning highlights the key role of predictions in mitigating the risk of multiyear droughts and designing adaptation procedures. Future climatic scenarios are usually generated by General Circulation Models (GCMs), which attempt to

simulate future climate variables under given scenarios of $CO_2$ emissions. Recently, the World Climate Research Programme (WCRP) released the Coupled Model Intercomparison Project Phase 6 (CMIP6; Eyring et al. (2016)). Simulations from several models are available which deliver ensemble projections of future climate.

The reliability of rainfall simulated by GCMs has been analysed by several studies that focused primarily on the generation of climate models that preceded CMIP6 (CMIP5 and previous models; see Palerme et al., 2017; Koutroulis et al., 2016;

Aloysius et al., 2016; Kumar et al., 2014b; Sillmann et al., 2013). Most of these researches compared the climatic scenarios with historical datasets produced by reanalysis such as NCEP or ERA-Interim (Palerme et al., 2017; Kumar et al., 2014b; Sillmann et al., 2013). However, on the one hand, reanalysis may introduce biases in the reproduction of extreme events. On the other hand, the observation period of these datasets covers only the recent past and therefore may undermine a statistical assessment of performances.

The present study aims at inspecting the ability of selected CMIP6 models in simulating regional scale climate by focusing on meteorological drought occurrence. We first compare simulations with one of the longest daily rainfall records globally available: the Bologna rainfall series, whose observation period dates back to 1813. We decided to adopt as baseline an observed





record, instead of a reanalysis, to take advantage from an extraordinary long observation period and avoid any bias that may be potentially induced by spatial interpolation.

The purpose of our study is twofold: (i) to evaluate the ability of up-to-date GCMs in simulating the statistics of observed precipitation by focusing on multiyear meteorological droughts and (ii) to infer how precipitation and drought risk will change in the future. The paper is organized as follows: Section 2 provides detailed descriptions of the data used in this study. Section 3 and 4 describe the methods for reliability testing and future change assessment, respectively. Section 5 presents the results of the historical evaluation and future projections for both precipitation and drought risk. Section 6 discusses the results. Finally,

Section 7 summarises our conclusions.

## 2    Data

### 2.1    Observation data

Italy is one of the first countries that started to systematically collect meteorological observations. Meteorological instruments and a network of observations were developed by Galileo's scholars and operated in the 17th century already. The rainfall series

collected in Padua since 1725 is the longest daily record in the world, and five other rainfall stations have been continuously in operation – with few missing values - since the eighteenth century (Bologna, Milan, Rome, Palermo and Turin). Therefore, a data set of enormous value has been accumulated in Italy over the last three centuries (Brunetti et al., 2006).

Rainfall observation in Bologna at daily time scale dates back to 1714. Since 1813 the series is continuous. Brunetti et al. (2001) provided an interesting description of the history of the time series and the procedures that were applied to detect and

resolve apparent inconsistencies. The observatory was originally located in the center of the city. The raingauge was changed in 1857 and likely in 1900. After 1978 data were collected by another observatory in a nearby location. Brunetti et al. (2001) proposed monthly correction factors to resolve apparent underestimation of rainfall during 1813-1858, 1900-1928 and 1813-1978. These corrections are valid for monthly rainfall only.

The daily rainfall series observed in Bologna from 1813 to 2021 can be obtained from the European Climate Assessment-

Daily Database (ECA & D) (https://climexp.knmi.nl/). It is one of the longest daily rainfall records worldwide publicly available. These are original data as reported in the transcripts of the observatory, without any correction. For the purpose of the present analysis, we assume that the daily series is homogeneous. In view of the time window adopted by CMIP6 GCMs for the historical reconstruction, we limit our analysis of the Bologna time series from 1850 to 2014 (see section 2.2).

Fig. 2 shows the daily time series and the 10-year moving average, which oscillates between a minimum of 1.2 mm and a

maximum of 2.5 mm (more than twice the minimum). The above extreme values occurred in the 1820s and the decade ending in 1902, respectively. After 1950, a mild increasing trend is noticed, which mirrors a similar tendency that occurred from about 1830 to about 1890.





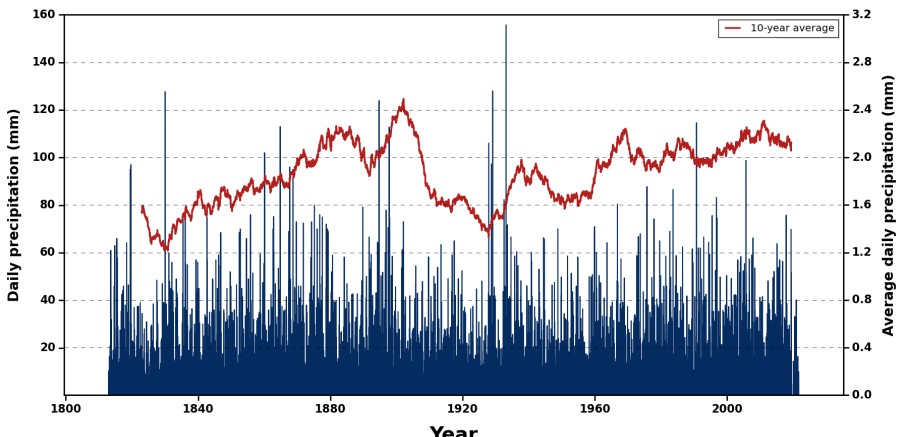

**Figure 2.** Daily rainfall records in Bologna, with 10-year moving average (red line)

## 2.2 General Circulation Model simulation

GCM simulations for both historical (1850-2014) and future (2015-2100) periods are publicly available from the Copernicus
database (https://cds.climate.copernicus.eu/). Future projections are obtained under the emission scenarios "Shared Socioeconomic Pathway" (SSP) 1-2.6 (SSP1-2.6), 2-4.5 (SSP2-4.5), and 5-8.5 (SSP5-8.5). These are the future emissions considered
by the Scenario Model Intercomparison Project (ScenarioMIP) (O'Neill et al., 2016) of CMIP6.

From each model an ensemble simulation is generated for different initial conditions, initialization methods, physics versions, and forcing datasets. Similarly to Grose et al. (2020) and Kim et al. (2020), we analyse only the first run of each
ensemble, that is identified by the acronym "r1i1p1f1", where "r", "i", "p" and "f" indicate initial conditions, initialization
method, physics version and forcing data set, respectively (see https://pcmdi.llnl.gov/CMIP6/Guide/dataUsers.html for more
details). We assume that analysing a single ensemble member per GCM model is sufficient to sample the ensemble for testing
model bias. The assumption above was recently proved to be tenable by Longmate et al. (2022).

Table 1 shows detailed information for 13 GCMs used in this study. We select all the models providing future simulations
for the three considered emission scenarios. The multi-model ensemble mean is the arithmetic average value of the 13 GCM
outputs. To compare the simulation with observed data, bilinear spatial interpolation is applied to estimate the model prediction
for Bologna depending on the four nearest GCM grid points. Given that GCM simulations start from 1850, we adopt the time
window 1850-2014 as our "historical period".

## 3 Reliability testing

The reliability of GCM simulations is evaluated by focusing on different temporal scales and both low and high rainfall, to
obtain a comprehensive picture of model performances.



**Table 1.** Description of 13 GCMs from CMIP6 used in this study.

| Model | Institution | Resolution | Grid (lon × lat) |
|---|---|---|---|
| CMCC-ESM2 | Fondazione Centro Euro-Mediterraneo sui Cambiamenti Climatici, Italy | $1.25° \times 0.94°$ | $288 \times 192$ |
| NorESM2-MM | Norwegian Climate Centre, Norway | $1.25° \times 0.94°$ | $288 \times 192$ |
| CanESM5 | Canadian Centre for Climate Modelling and Analysis, Environment and Climate Change, Victoria, Canada | $2.81° \times 2.81°$ | $128 \times 64$ |
| INM-CM5-0 | Institute for Numerical Mathematics, Russian Academy of Science, Russia | $2.00° \times 1.50°$ | $180 \times 120$ |
| INM-CM4-8 | Institute for Numerical Mathematics, Russian Academy of Science, Russia | $2.00° \times 1.50°$ | $180 \times 120$ |
| MPI-ESM1-2-LR | Max Planck Institute for Meteorology, Germany | $1.875° \times 1.875°$ | $192 \times 96$ |
| MIROC6 | Japan Agency for Marine-Earth Science and Technology, Japan | $1.40° \times 1.40°$ | $256 \times 128$ |
| EC-Earth3-Veg-LR | Consortium of European Research Institution and Researchers, Europe | $1.125° \times 1.125°$ | $320 \times 160$ |
| GFDL-ESM4 | Geophysical Fluid Dynamics Laboratory, USA | $1.25° \times 1.00°$ | $360 \times 180$ |
| MRI-ESM2-0 | Meteorological Research Institute, Japan | $1.125° \times 1.125°$ | $320 \times 160$ |
| ACCESS-CM2 | Commonwealth Scientific and Industrial Research Organization- Australian Research Council Centre of Excellence for Climate System Science, Australia | $1.875° \times 1.250°$ | $192 \times 144$ |
| FGOALS-g3 | Institute of Atmospheric Physics, Chinese Academy of Sciences, China | $2.00° \times 2.25°$ | $180 \times 80$ |
| IPSL-CM6A-LR | Institut Pierre Simon Laplace, Paris, France | $2.50° \times 1.25°$ | $144 \times 143$ |

## 3.1 Annual rainfall

To evaluate the performance of each of the considered CMIP6 GCMs in reproducing the behavior of annual rainfall we evaluate their capability to reproduce the observed hyetograph and its statistics during the historical period. The goodness of the hyetograph simulation is evaluated by computing the modified Kling-Gupta efficiency (KGE) (Kling et al., 2012), the mean absolute relative error (MARE), and the Nash-Sutcliffe Efficiency (NSE) (Nash and Sutcliffe, 1970). GCM historical simulations are divided into two periods: long-past (1850-1949) and near-past (1950-2014).

KGE is computed by combining three statistical parameters: correlation coefficient ($r$), bias ratio ($\beta$), and variability ratio ($\gamma$)):

$$KGE = 1 - \sqrt{(r-1)^2 + (\beta-1)^2 + (\gamma-1)^2} \tag{1}$$

with

$$r = \frac{\sum_{i=1}^{n}(obs_i - \mu_{obs})(GCM_i - \mu_{GCM})}{\sqrt{\sum_{i=1}^{n}(obs_i - \mu_{obs})^2}\sqrt{\sum_{i=1}^{n}(GCM_i - \mu_{GCM})^2}}, \tag{2}$$

$$\beta = \frac{\mu_{GCM}}{\mu_{obs}}, \tag{3}$$





$$\gamma = \frac{\sigma_{GCM}/\mu_{GCM}}{\sigma_{obs}/\mu_{obs}}. \tag{4}$$

In eq. 1-4, $obs_i$ and $GCM_i$ indicate annual rainfall at year $i$ given by historical observations and the considered GCM simulation, respectively, $\mu_{obs}$ and $\mu_{GCM}$ represent the mean value along the considered period of observed data and GCM, respectively, and $\sigma_{obs}$ and $\sigma_{GCM}$ indicate the corresponding standard deviation.

110     MARE and NSE are computed as follows:

$$MARE = \frac{\sum_{i=1}^{n} \mid obs_i - GCM_i \mid}{\sum_{i=1}^{n} obs_i}, \tag{5}$$

$$NSE = 1 - \frac{\sum_{i=1}^{n} (obs_i - GCM_i)^2}{\sum_{i=1}^{n} (obs_i - \mu_{obs})^2}. \tag{6}$$

As for the hyetograph statistics, we first make a visual comparison of the sample probability density of observed and simulated annual rainfall. Moreover, we compare the lag 1 autocorrelation coefficient of historical data versus each simulation.

### 3.2    Mean monthly and seasonal rainfall

The goodness of the simulation of mean monthly and seasonal rainfall is evaluated by a graphical comparison with the observed values and the Taylor diagram (Taylor, 2001). The latter is widely used to summarise how accurately a model simulates an observed record. It integrates three statistical metrics: correlation (R), centered root-mean-square error (CRMSE) and ratio of spatial standard deviation (SD) in a single diagram. These statistics provide a quick summary of the correspondence between the modeled and observed mean monthly and seasonal rainfall, which is particularly useful in assessing the relative merits and overall performance of climate models (Rivera and Arnould, 2020; Dong and Dong, 2021; Yazdandoost et al., 2021). The Taylor diagram is herein used to assess the goodness of the fit of the (a) mean monthly rainfall, (b) March-April-May (MAM) and (c) September-October-November (SON) mean rainfall.

### 3.3    Extreme rainfall

The climate extreme indexes suggested by the Expert Team on Climate Change Detection and Indices (ETCCDI) (Zhang et al., 2011) have been widely used in the evaluation and projection of climate extremes (Faye and Akinsanola, 2022; Xu et al., 2019; Schoof and Robeson, 2016; Casanueva et al., 2014). Table 2 describes the eight indexes that are herein used to evaluate the change of the rainfall extremes. More detailed information on the indexes can be found on the ETCCDI website (http://etccdi.pacificclimate.org/indices.shtml).

We compute the relative root-mean-square error ($RMSE^{'}$) (Ge et al., 2019, 2021) of the simulation by each model of each index as follows. First, the root-mean-square error ($RMSE$) is calculated for each GCM in reproducing a given index:

$$RMSE = \sqrt{(X - Y)^2} \tag{7}$$





**Table 2.** Description of extreme indices used in this study.

| Label | Description | Index definition | Units |
|---|---|---|---|
| CWD | Consecutive wet days | Maximum number of consecutive days when rainfall $\geq$ 1 mm. | days |
| CDD | Consecutive dry days | Maximum number of consecutive days when rainfall $<$ 1 mm. | days |
| R10mm | Heavy rainfall days | Count of days when rainfall $\geq$ 10 mm. | days |
| R20mm | Very heavy rainfall days | Count of days when rainfall $\geq$ 20 mm. | days |
| Rx1day | Maximum daily rainfall | Maximum daily rainfall. | mm |
| Rx5day | Maximum 5-day rainfall | Maximum 5-day rainfall. | mm |
| R95p | Very wet days | Count of days when rainfall $\geq$ 95th percentile. | days |
| R95pTOT | Very wet days rainfall | Total rainfall from days $\geq$ 95th percentile. | mm |

where X is a given extreme rainfall index simulated by the considered model and Y is the corresponding observed value. A

standardized value $RMSE'$ of the $RMSE$ of each model is then computed as follows:

$$RMSE' = \frac{(RMSE - RMSE_{Median})}{RMSE_{Median}} \tag{8}$$

where $RMSE_{Median}$ is the median RMSE of all models. A negative $RMSE'$ indicates a better model performance than half of the models. For instance, a value of $RMSE' = -0.4$ means that the model $RMSE$ for the given index is 40% smaller than the median value across all models (Gleckler et al., 2008).

**3.4    Meteorological droughts**

To test GCM's reliability in simulating multiyear meteorological droughts we apply run theory (Yevjevich, 1967) to annual rainfall to characterise drought events in terms of drought frequency, duration, severity and intensity. Run theory is one of the most effective approaches for drought identification and has been applied in several areas worldwide (Ho et al., 2021; Wu et al., 2020; Mishra et al., 2009).

In detail, the long term mean rainfall $R_{LT}$ is adopted as the threshold to identify positive or negative runs (see Fig. 3). If rainfall in a given year is lower than an assigned threshold $T_{lower} < R_{LT}$ a negative run is started which ends in the year when the rainfall is higher than $R_{LT}$. If the interval between two negative runs is only one year and rainfall in that year is less than a selected threshold $T_{upper} > R_{LT}$, then these two runs are combined into one drought. Finally, only runs which have a duration of no less than 3 years are determined as multiyear drought events. Here, the thresholds $T_{upper}$ and $T_{lower}$ are defined as 20%

more and 10% less than $R_{LT}$, respectively.

Once a multiyear drought has been identified, drought duration is the time span between the start and the end of the event, and drought severity is computed as the cumulative rainfall deficit with respect to $R_{LT}$ during drought duration divided by the mean rainfall, and drought intensity is computed as the ratio between drought severity and duration. We also estimate the maximum deficit of the drought event, namely, the largest difference between annual rainfall and $R_{LT}$ during the event. Finally,



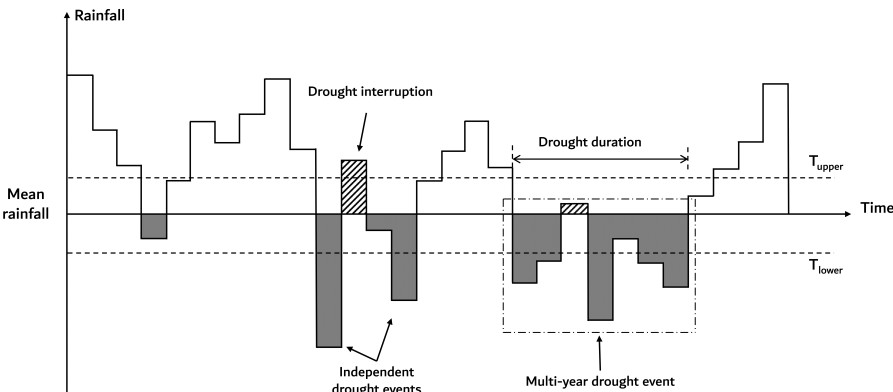

**Figure 3.** Identification of multiyear drought events and characteristics by using run theory.

drought frequency is computed as the ratio between the number of drought events that have been identified and the length of the observation period.

## 4 Future climate change assessment

Statistics of future projections of annual and seasonal rainfall of the 13 considered GCMs under the three considered scenarios are compared with observed and simulated statistics of the historical period to evaluate future climate change in the Bologna region. For a detailed comparison of seasonal rainfall, the future time horizon is divided into near-future (2030-2059) and long-future (2070-2099) time windows. The multi-model median and the 25th–75th percentiles of the projections given by the GCMs are considered for each scenario to represent the associated ensemble uncertainty.

## 5 Results

### 5.1 Reliability of historical data simulation

#### 5.1.1 Annual rainfall

Fig. 4 and Fig. 5 show a comparison of simulated versus observed hyetographs for each of the 13 CMIP6 GCMs thus providing a visual assessment of their reliability for long-past and near-past historical period, respectively. Some models (e.g. CanESM5 and MPI-ESM1-2-LR) generally underestimate the annual rainfall for both periods while other models (e.g. ACCESS-CM2 and GFDL-ESM4) end up with a prevailing overestimation. Some other models perform differently in the two considered



historical time periods. In INM-CM5-0 and NorESM2-MM, for example, long-past rainfall is relatively better reproduced than near-past rainfall while the opposite result is found for INM-CM4-8, which can better simulate near-past rainfall. The multi-model ensemble (MME), namely the mean value of the 13 GCMs, is characterized by a much lower variability than the historical data and therefore does not provide any improvement with respect to each individual GCM.

Fig. 6 shows the obtained goodness of fit indexes. Good model performances correspond to a high positive correlation
coefficient (R), high Kling-Gupta efficiency (KGE), high Nash-Sutcliffe efficiency (NSE), and a small mean absolute relative error (MARE). The metrics are calculated for long-past, near-past, and total historical period 1850-2014.

For both long-past and near-past periods, 4 of the models (GFDL-ESM4, CMCC-ESM2, FGOALS-g3, and INM-CM5-0) and the MME show a positive R while MIROC6, ACCESS-CM2, and NorESM2-MM are characterised by negative R in both periods. Some other models perform differently in each time period. INM-CM4-8 and CanESM5 show a positive R in near-
past and negative in long-past. It is interesting to note that INM-CM4-8 obtains the highest R in near-past. An opposite result is obtained for MPI-ESM1-2-LR, EC-Earth3-Veg-LR and MRI-ESM2-0 with the latter showing the highest negative value in near-past. INM-CM5-0 exhibits the highest R in the long-past but shows no significant correlation in the near-past while FGOALS-g3 obtains the highest R in the near-past. In addition, the IPSL-CM6A-LR exhibits the highest negative value in simulating long-past rainfall. In general, the positive correlation is low for all models.

The MME delivers the lowest MARE in each period and performs similarly for both near-past and long-past. ACCESS-CM2, GFDL-ESM4, and MPI-ESM1-2-LR show a relatively higher MARE for both periods while ACCESS-CM2 shows the highest MARE in each period. The other models exhibit a relatively smaller MARE and perform slightly different in each period.

As for KGE, 5 of 13 models and the MME show negative KGE values for both long-past and near-past. The ACCESS-CM2 and the MME show the highest negative value for long-past and near-past, respectively. Furthermore, CanESM5 and IPSL-
CM6A-LR exhibit a high negative value for long-past while the values are near to zero for the near-past. Only FGOALS-g3, GFDL-ESM4, and CMCC-ESM2 depict relatively higher KGE values for both periods while the INM-CM5-0 and INM-CM4-8 show the highest value for long-past and near-past, respectively.

All models and the ensemble show negative values of NSE which indicate an overall poor ability in replicating historical annual rainfall. ACCESS-CM2, MPI-ESM2-2-LR, and GFDL-ESM4 depict relatively smaller values for both periods with the
ACCESS-CM2 showing the smallest value in each period. The multi-model ensemble and FGOALS-g3 exhibit a higher NSE with a value near zero, which presents a similar performance to the mean of observation data.

In general, CMIP6 GCMs do not satisfactorily reproduce the historical sequence of annual rainfall, with the less satisfactory performances exhibited by ACCESS-CM2, MRI-ESM2-0, and EC-Earth3-Veg-LR. FGOALS-g3, GFDL-ESM4 and CMCC-ESM2 perform relatively better for both periods while INM-CM5-0 and INM-CM4-8 show the best ability in simulating long-
past and near-past annual rainfall, respectively. Interestingly, all the models show a better performance during one separate period than the total historical period. Compared to every single model, the MME shows no obvious better capability in replicating the long sequence of annual rainfall.

We further investigate the ability of GCMs in capturing the sample probability density for annual rainfall in the whole historical period. Fig. 7 shows that the majority of models perform worse in capturing heavier rainfall than lighter rainfall,







**Figure 4.** Long-past historical observed and simulated by CMIP6 models annual rainfall time series.

with the opposite result occurring in EC-Earth3-Veg-LR, MIROC6 and MRI-ESM2-0. Only ACCESS-CM2 and GFDL-ESM4 overestimate both high and low rainfall. It is noted that MME exhibits lower variability.

Table 3 displays the lag-1 autocorrelation coefficient for each model simulation. It is interesting to note that correlation of observed data is slightly higher than all the models, therefore highlighting possible model weakness in simulating long runs.





**Figure 5.** Near-past historical observed and simulated by CMIP6 models annual rainfall time series.

**Table 3.** Lag-1 autocorrelation coefficient between observed data and each historical simulation.

| Label | OBS | MME | 1 | 2 | 3 | 4 | 5 | 6 | 7 | 8 | 9 | 10 | 11 | 12 | 13 |
|-------|-----|-----|---|---|---|---|---|---|---|---|---|----|----|----|----|
| Lag-1 | 0.226 | 0.052 | −0.012 | 0.122 | 0.036 | 0.173 | 0.127 | 0.099 | 0.085 | −0.084 | 0.018 | 0.086 | 0.067 | 0.081 | −0.052 |

OBS is observation data, MME is multi-model ensemble mean and numbers indicate models: 1. ACCESS-CM2; 2. CMCC-ESM2; 3.CanESM5; 4.EC-Earth3-Veg-LR; 5. FGOALS-g3; 6. GFDL-ESM4; 7. INM-CM4-8; 8. INM-CM5-0; 9. IPSL-CM6A-LR; 10. MIROC6; 11. MPI-ESM1-2-LR; 12. MRI-ESM2-0; 13. NorESM2-MM.





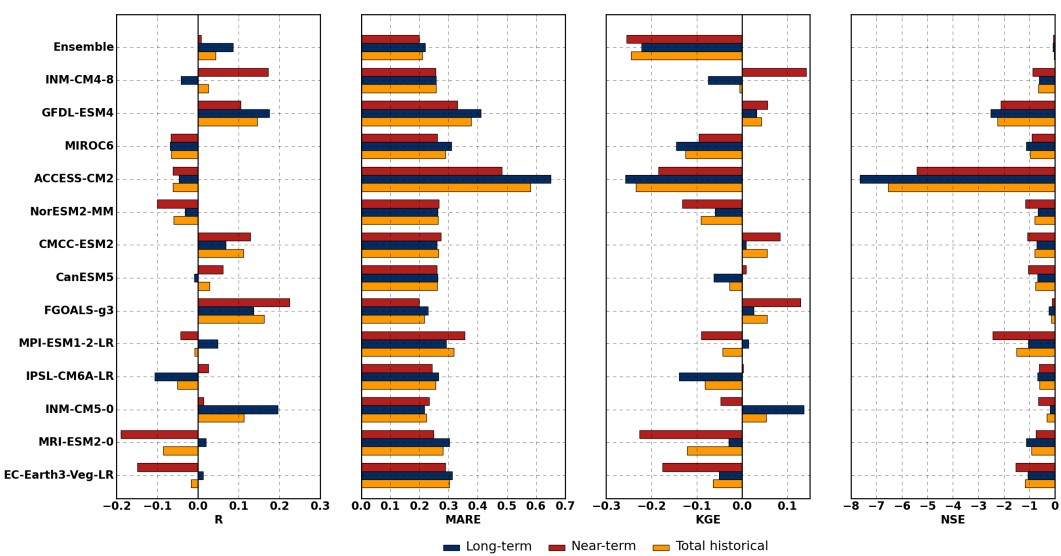

**Figure 6.** Correlation coefficient (R), Mean Absolute Relative Error (MARE), Kling-Gupta efficiency (KGE), Nash-Sutcliffe efficiency (NSE) of CMIP6 GCMs and ensemble simulation.





**Figure 7.** Comparison of sample probability density for observations and GCM historical simulations.




### 5.1.2 Mean monthly rainfall

Fig. 8 shows the graphical comparison between observed and simulated mean monthly rainfall for the whole historical period.
Most of the models adequately replicate the seasonal rainfall pattern except IPSL-CM6A-LR and FGOALS-g3. However,
ACCESS-CM2 significantly overestimates the rainfall for all months. Moreover, all models but MPI-ESM1-2-LR, CMCC-
ESM2, NorESM2-MM, and CanESM5 overestimate the rainfall in December-January-February. Conversely, all models but
ACCESS-CM2, GFDL-ESM4, EC-Earth3-Veg-LR, and MRI-ESM2-0 underestimate the June-July-August and/or September-
October-November rainfall. For the March-April-May rainfall, half of the models exhibit overestimation. In addition, the MME
satisfactorily reproduces the annual cycle of rainfall, especially the March-April-May rainfall while slightly underestimating
the June-July-August and September-October-November rainfall and overestimating the December-January-February rainfall.

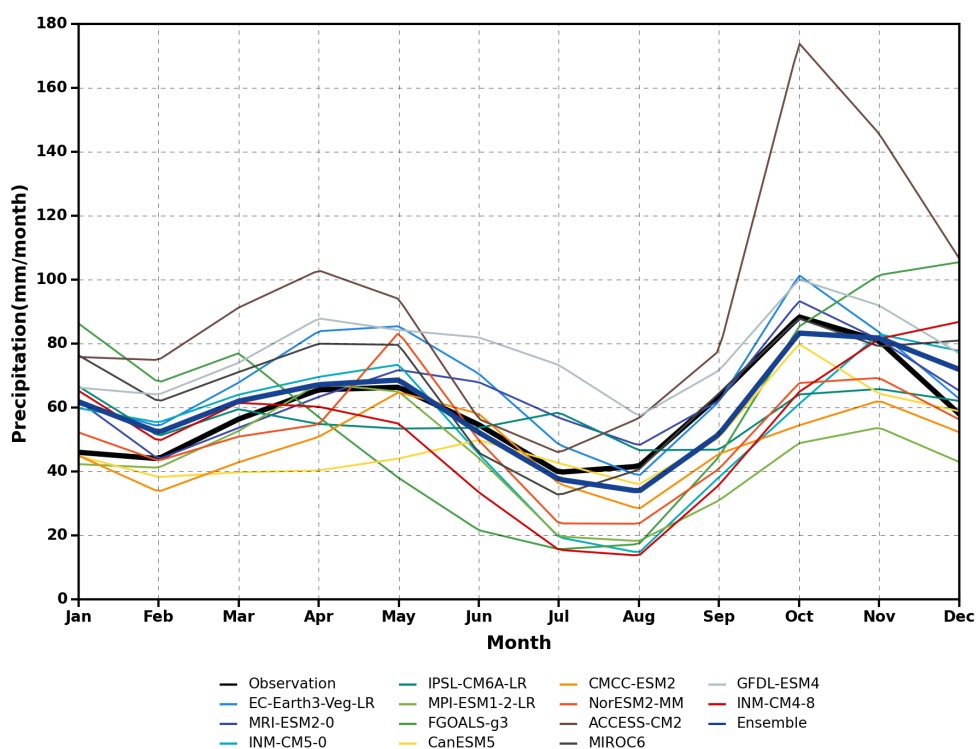

**Figure 8.** Annual cycle of historical (1850-2014) mean monthly rainfall (mm/month) of MME and CMIP6 models against observation data.

       Fig. 9 shows the Taylor diagram for each GCM and the MME in simulating the seasonal rainfall pattern. It confirms that 6
of the models and the MME adequately replicate the mean monthly rainfall with relatively high R, low CRMSE, and low SD.
In particular, EC-Earth3-Veg-LR and MRI-ESM2-0. IPSL-CM6A-LR and FGOALS-g3 display relatively poor performance.
However, a significant improvement is exhibited in reproducing the March-April-May and September-October-November rain-
fall for these two models, especially the FGOALS-g3 that can simulate the September-October-November rainfall best. Most




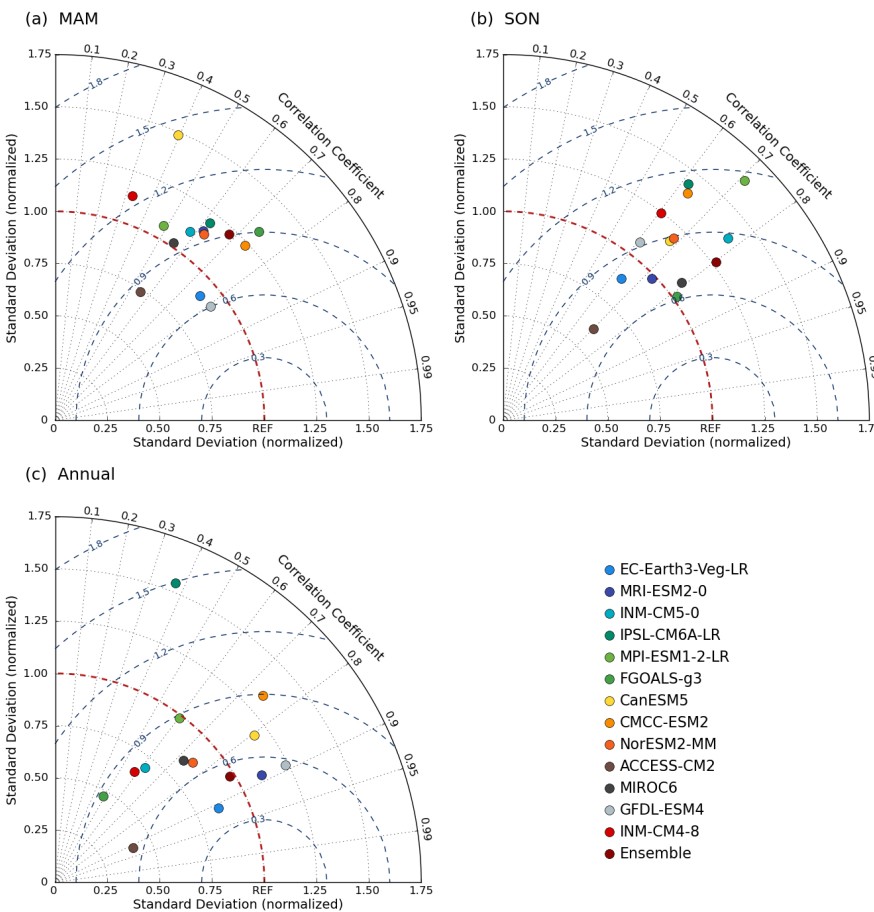

**Figure 9.** Taylor diagram of (a) March-April-May (MAM), (b) September-October-November (SON), and (c) Annual cycle of historical (1850-2014) mean monthly rainfall for the CMIP6 models and their MME.

of the models depict a higher R but larger bias in September-October-November than March-April-May. In general, the EC-Earth3-Veg-LR shows a satisfactory performance in simulating both annual cycle and March-April-May rainfall while GFDL-

ESM4 and FGOALS-g3 can better reproduce the March-April-May and September-October-November rainfall, respectively. Although R is slightly lower than the EC-Earth3-Veg-LR, the MME performs satisfactorily in simulating the annual cycle of rainfall but shows no significant advance over the single models in reproducing March-April-May and September-October-November rainfall.





### 5.1.3 Extreme rainfall

The $RMSE'$ of individual model in simulating rainfall extremes are summarized in Fig. 10. MRI-ESM2-0 performs fairly well with all negative $RMSE'$ values for different indices. Two models, EC-Earth3-Veg-LR and GFDL-ESM4, provide acceptable performance with mainly negative $RMSE'$ values for different indices. The $RMSE'$ of the multi-model mean is shown in the last column of Fig. 10. The ensemble mean performs better than any individual model for all indices.

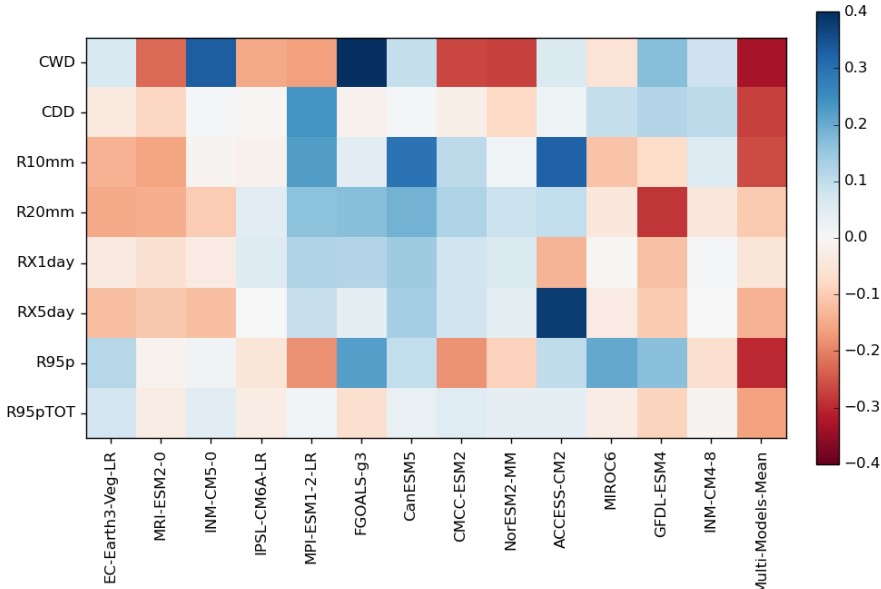

**Figure 10.** Portrait diagram of $RMSE'$ for different extreme rainfall indices simulated by CMIP6 models.

### 5.1.4 Meteorological droughts

Table 4 shows the characteristics of multiyear droughts derived by using run theory for both observations and each model simulation for both the whole historical and whole future periods, the latter for SSP2.6 and SSP8.5 scenarios.

The results highlight that FGOALS-g3, INM-CM5-0, MIROC6, MPI-ESM2-0 and NorESM2-MM simulate drought frequency (DF) fairly well. Notably, all models fail to replicate the drought duration (DD), drought intensity (DI) and maximum deficit (MD). For instance, IPSL-CM6A-LR and INM-CM4-8 show the best performance in simulating DD, which, however,

is underestimated by about 10% by the best simulation. Although MPI-ESM1-2-LR presents the highest value of DI and MD against all the models, marked underestimation with respect to the observations still occurs. MME displays relatively good performance in terms of DF but still underestimates DD, DI and MD. In detail, the MME DD is underestimated by about 17%, while DI and MD for observations are even nearly 34% and 39% higher than MME, respectively.





**Table 4.** Statistics for multiyear meteorological droughts exhibited by observed data (1850-2014) and reproduced by models for the historical (1850-2014) and future (2015-2100) periods under the two considered scenarios.

| Models | Drought Frequency | | | Drought Duration | | | Drought Intensity | | | Max Deficit | | |
|---|---|---|---|---|---|---|---|---|---|---|---|---|
| | HIS | SSP2.6 | SSP8.5 | HIS | SSP2.6 | SSP8.5 | HIS | SSP2.6 | SSP8.5 | HIS | SSP2.6 | SSP8.5 |
| OBS | 0.055 | - | - | 4.80 | - | - | 0.205 | - | - | 36.70% | - | - |
| MME | 0.053 | 0.056 | 0.050 | 3.96 | 4.02 | 4.75 | 0.153 | 0.168 | 0.184 | 26.40% | 27.87% | 32.24% |
| 1 | 0.030 | 0.070 | 0.058 | 4.20 | 3.83 | 4.20 | 0.134 | 0.184 | 0.202 | 27.05% | 33.23% | 39.55% |
| 2 | 0.042 | 0.035 | 0.035 | 3.86 | 4.00 | 4.00 | 0.167 | 0.154 | 0.192 | 23.13% | 27.05% | 31.84% |
| 3 | 0.042 | 0.047 | 0.058 | 4.14 | 3.75 | 4.20 | 0.155 | 0.188 | 0.210 | 29.89% | 28.32% | 36.45% |
| 4 | 0.079 | 0.081 | 0.058 | 4.23 | 3.71 | 4.60 | 0.140 | 0.171 | 0.203 | 24.60% | 28.18% | 34.16% |
| 5 | 0.055 | 0.035 | 0.035 | 4.11 | 3.00 | 3.67 | 0.130 | 0.110 | 0.091 | 22.03% | 18.29% | 19.21% |
| 6 | 0.048 | 0.035 | 0.047 | 3.63 | 6.33 | 4.50 | 0.162 | 0.152 | 0.173 | 28.76% | 25.12% | 26.90% |
| 7 | 0.048 | 0.070 | 0.070 | 4.25 | 4.00 | 3.83 | 0.167 | 0.155 | 0.177 | 27.27% | 23.55% | 29.40% |
| 8 | 0.055 | 0.058 | 0.058 | 4.33 | 3.60 | 4.60 | 0.131 | 0.189 | 0.189 | 21.66% | 30.67% | 34.00% |
| 9 | 0.048 | 0.081 | 0.023 | 4.25 | 3.57 | 9.00 | 0.132 | 0.177 | 0.246 | 26.86% | 28.29% | 48.37% |
| 10 | 0.055 | 0.070 | 0.047 | 3.44 | 3.67 | 5.75 | 0.163 | 0.147 | 0.138 | 26.55% | 24.34% | 21.95% |
| 11 | 0.079 | 0.047 | 0.047 | 3.54 | 3.75 | 5.00 | 0.173 | 0.226 | 0.184 | 29.03% | 37.97% | 34.04% |
| 12 | 0.055 | 0.035 | 0.058 | 3.89 | 5.67 | 4.40 | 0.173 | 0.161 | 0.191 | 28.75% | 26.34% | 28.02% |
| 13 | 0.055 | 0.070 | 0.058 | 3.56 | 3.33 | 4.00 | 0.158 | 0.165 | 0.194 | 27.63% | 30.93% | 35.23% |

The unit of drought frequency is (times/year) and the unit of drought duration is (years). OBS and HIS are observed data and historical simulation. MME is multi-model ensemble mean and different numbers indicate different models: 1. ACCESS-CM2; 2. CMCC-ESM2; 3.CanESM5; 4.EC-Earth3-Veg-LR; 5. FGOALS-g3; 6. GFDL-ESM4; 7. INM-CM4-8; 8. INM-CM5-0; 9. IPSL-CM6A-LR; 10. MIROC6; 11. MPI-ESM1-2-LR; 12. MRI-ESM2-0; 13. NorESM2-MM.

In general, the multi-model ensemble can satisfactorily simulate the frequency of the multiyear meteorological droughts while significantly underestimating drought duration, intensity and maximum deficit.

## 5.2 Future changes

### 5.2.1 Changes in average rainfall

Fig. 11 shows the future projections of annual rainfall for three different scenarios (SSP1-2.6, SSP2-4.5 and SSP5-8.5), compared with the MME simulation of the historical period (1850-2014). Future annual rainfall shows only a slight decrease for the SSP5-8.5 scenario while fluctuating with close to stationary variability with respect to the historical period. To inspect the temporal progress of changes, the annual cycle of rainfall is considered and the future period is divided into near-future (2030-2059) and long-future (2070-2099) related to the present-day simulation (1985-2014). Fig. 12 shows an overall decrease in monthly rainfall under all the scenarios in each future period. Under the SSP2.6 scenario, the March-April-May rainfall in the long-future is expected to be less than in the near-future period. Conversely, the rainfall during the September-



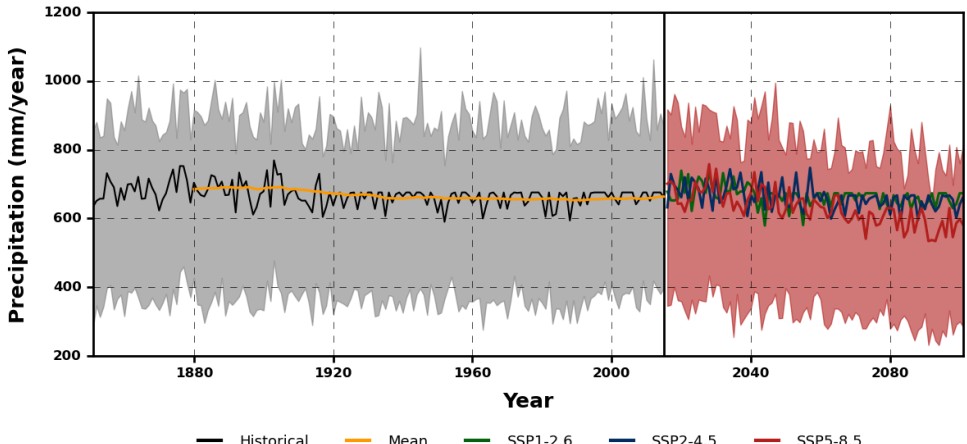

**Figure 11.** Time series of annual rainfall for both historical simulation and future projections (mm/year). Historical (black), 30-year mean of historical (yellow), future MME simulations for SSP1-2.6 (green), SSP2-4.5 (blue), and SSP5-8.5 (red). Uncertainty range for historical simulation (grey shading) and future projection under SSP5-8.5 (red shading) are shown in the 25th and 75th percentiles of the ensemble.

October-November season in the long-future will be higher than in the near-future. Moreover, rainfall may be more likely to be concentrated in November rather than in October in both periods. This change in rainfall pattern also occurs in the near-future period under the SSP4.5 scenario, but the overall monthly rainfall shows no significant difference between periods. Under the SSP8.5 scenario, the March-April-May and June-July-August rainfall in the long-future will be significantly less than in the near-future and the rainfall will be more concentrated in the SON and DJF season.

### 5.2.2 Future meteorological drought changes

Table 4 reports future drought changes for SSP2.6 and SSP8.5 future scenarios, which represent low emission pathway and high emission pathway, respectively. Corresponding statistics for observed data (whole 1850-2014 observations) and model simulations under historical period (1850-2014) are also shown. For drought frequency (DF), the future changes with respect to historical simulations are not remarkable. Note that this statistic was fairly well reproduced by the simulation of models under historical period with respect to observed DF. The MME also shows no significant change, with DF slightly increasing under SSP2.6 but declining under SSP8.5 compared to historical simulation.

The future changes are much more significant for drought duration (DD), drought intensity (DI) and maximum deficit (MD), which, however, were underestimated by models when simulating the historical period. The DD of most of the models is longer under SSP8.5, while being shorter under SSP2.6, with respect to historical simulation. Nearly all the models show a marked increase in DI for at least one future scenario and the majority of models show a more intense drought under SSP8.5 than SSP2.6 except FGOALS-g3 and MIROC6, which show a continuous decline in both future scenarios. The changes in MD are similar to DI. Additionally, the MME shows a continuous increase for DD, DI and MD with a relatively larger increase under

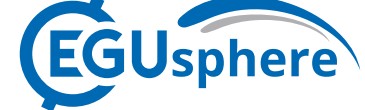

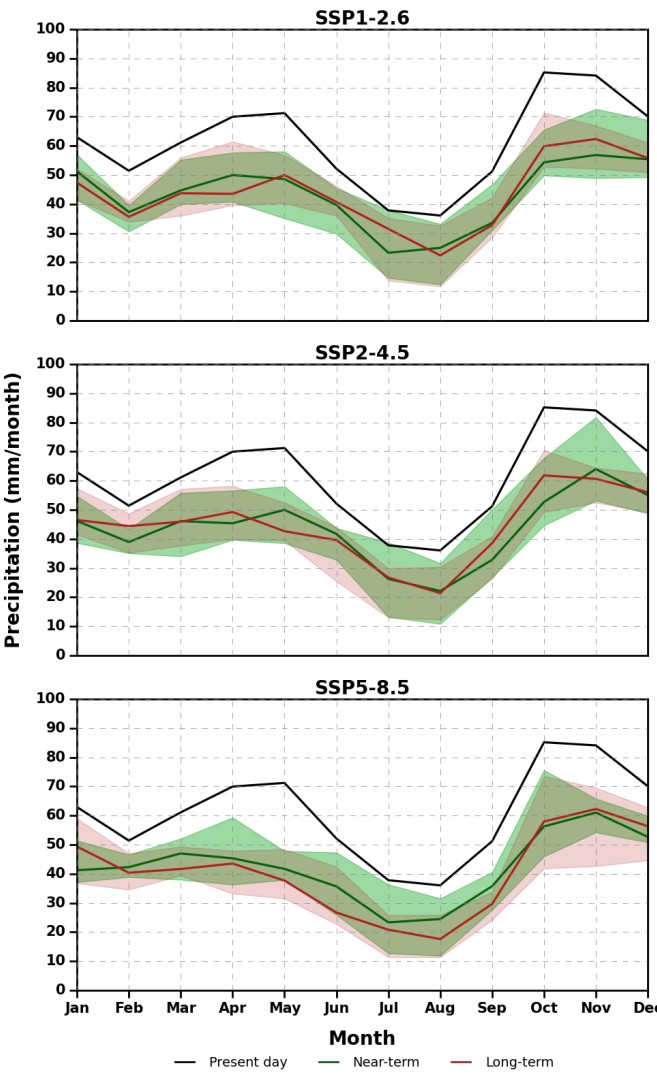

**Figure 12.** Monthly mean rainfall (mm/month) for multi models in present-day (1985-2014), near-future (2030-2059) and long-future (2070-2099) under SSP1-2.6, SSP 2-4.5 and SSP5-8.5 scenarios. Thick lines indicate multi-model medians while shading indicates the 25th–75th percentile model ranges.

SSP8.5. The DD will only slightly increase under SSP2.6 while resulting 19.9% longer under SSP8.5. Similar changes occur in MD, which will increase by 22.1% under SSP8.5. The MME shows that DI will increase by 9.8% under SSP2.6, while a much

larger increase of 20.3% under SSP8.5. We should note that the climate models do not show consistent predictions for future changes in statistics of multiyear droughts, but they depict overall drier conditions in the high-emission scenario compared to both low-emission scenario and historical simulation.



In general, few models only predict the worst meteorological drought statistics during 2015-2100 with respect to 1850-2014 observations, and MME does not resolve the problem as it delivers a less conservative prediction with respect to past

occurrences of multiyear droughts.

## 6  Discussion

As the results show, the GCM cannot satisfactorily reproduce the chronological annual rainfall series and the performance of the model will depend on the selected window of the historical period. However, the multi-model ensemble can simulate the monthly rainfall pattern and extreme rainfall fairly well, as well as drought frequency. Conversely, drought duration, intensity

and maximum deficit are underestimated by models.

Therefore, one may expect that future projections for different emission scenarios provided by GCM models may deliver useful predictions of rainfall statistics to assess the impact of climate change. However, one should note that the projected changes in average rainfall are not remarkable: only for SSP8.5, a decrease is predicted of about 10% with respect to the historical period. The seasonal rainfall pattern is not predicted to change significantly.

For multiyear meteorological droughts, that is our main focus, we pointed out above that the multi-model ensemble can satisfactorily simulate their frequency while significantly underestimating the duration, intensity and maximum deficit. Additionally, the duration, intensity and maximum deficit predicted by models in the future are generally less critical than what was observed in the past. Such evidence suggests carefully considering the information supporting the technical design of climate change adaptation policies for droughts. For the considered case of the Bologna region, extreme drought estimation carried out

by statistical analysis of historical records under the assumption of stationarity delivers a more critical future prediction with respect to climate model simulations.

## 7  Conclusions

The present study refers to the region of Bologna, where the availability of a 209-year-long daily rainfall series allows us to make a unique assessment of GCM reliability and their predicted changes in rainfall. Our results confirm the poor capability

of GCM to simulate the chronological order of observations, therefore highlighting that their uncertainty should be carefully considered, which is not effectively accounted for by drawing ensemble simulations.

GCMs perform fairly well in terms of predicted statistics, with uncertainty that is however expected to depend on the considered design variable. GCMs' predictions for the future generally deliver a worse picture with respect to present day simulations, but our results suggest carefully considering the impact of uncertainty when designing climate change adaptation

policies. For some situations, classical engineering methods for critical event estimation under the assumption of stationarity may turn out to be more precautionary. Therefore, rigorous use and comprehensive interpretation of the available information are needed to avoid mismanagement, by also taking into account that the impact of multiyear meteorological droughts is likely to be exacerbated by further pressure on water resources due to increasing population and water demand.





*Data availability.*  The historical rainfall series observed in Bologna can be obtained from https://climexp.knmi.nl/. All the CMIP6 GCM
outputs are publicly available from the Copernicus Data Store, https://cds.climate.copernicus.eu/.

*Author contributions.*  AM proposed the main research question and supervised the work. RG made the computational analysis, elaborated
additional research ideas and prepared the manuscript.

*Competing interests.*  The authors declare no competing interests.

*Acknowledgements.*  RG was supported by the China Scholarship Council (CSC) Scholarship NO.202106060061. We thank Dr. Zhiqi Yang
and Zhanwei Liu for the helpful discussions.



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
