# Peer review of "Historical rainfall data in Northern Italy predict larger meteorological drought hazard than climate projections"

_EGUsphere, 2022_

## Author Response (AR1)

**Historical rainfall data in Northern Italy predict larger meteorological drought hazard than climate projections**

Rui Guo and Alberto Montanari

Department of Civil, Chemical, Environmental and Materials Engineering (DICAM), University of Bologna, Bologna, Italy

Correspondence: Rui Guo (rui.guo2@unibo.it)

We are grateful to the Editor for managing the review process of our paper. We are also grateful to the reviewers for their positive comments and critical suggestions, which helped us to improve the manuscript. The review comments and the Editor's comments were indeed very constructive.

Within this rebuttal document we describe how the remarks by the Reviewers and the Editor were addressed. Quotes from the reviews are reported in italic. We include here below selected quotes from the revised manuscript to better explain how the related concerns were addressed. Quotes from the revised manuscript are highlighted in blue.

**Reply to Anonymous Reviewer #1**

We thank the reviewer for the thorough and helpful review of our manuscript and the generally positive feedback. Here below we explain how the comments of the reviewer had been addressed. Comments are quoted in italic.

*1) Maybe the following lines (i.e., lines 278-280) could be somewhat extended for including as much information as possible from the analysis outputs: "In general, few models only predict the worst meteorological drought statistics during 2015-2100 with respect to 1850-2014 observations, and MME does not resolve the problem as it delivers a less conservative prediction with respect to past occurrences of multiyear droughts". For instance, the "few models" and the "worst meteorological drought statistics" could be reported. Discussing the related analysis outputs in greater detail would be beneficial, to my view, as they consist one of the most interesting parts of the paper.*

We agree with the reviewer. We included an additional analysis to correct the projection bias and the corresponding results for drought analyse has been added. According to the suggestion of the reviewer, these have been discussed in detail in the section 6.2.2 as follows:

"The future changes of DF with respect to historical simulations are not remarkable. DD, DF and MD are generally underestimated with respect to historical data.

In fact, the values of DD, DI and MD for MME under SSP2.6 and SSP8.5 are both lower with respect to observations. Moreover, DD of all models but GFDL-ESM4 is shorter under SSP2.6 with respect to observed data. Only the MME and models

CMCC-ESM2, MIROC6 and MPI-ESM1-2 show a consistent increase of DD when turning from historical simulation to SSP2.6 and SSP8.5. Nearly all models show an increase in DI with respect to historical simulation for at least one future scenario and all models but FGOALS-g3 and MIROC6 show a more intense drought under SSP8.5 than SSP2.6. Changes in MD are similar to DI. The above considerations show that historical data depict a worse future in terms of multiyear droughts with respect to simulations before QDM.

Table 4 presents the corresponding future drought changes after QDM. The good performance of MME in terms of DF isconfirmed. However, more models exhibit less DF compared to observations under SSP8.5. For DD, QDM results in a general deterioration of performances in terms of underestimation. For DI and MD, similar values to observed data are reached by MME under SSP8.5 only, but with large variability among models. In general, QDM improves MME performances in but large variability among models is not resolved. Moreover, the expectation of increased drought risk in the future with respect to historical observations is not confirmed even after QDM and the application of the worst emission scenario."

We hope that this description addressed the reviewer's concerns.

*2) Additional recommendations for future research (aside from the general directions already provided in the paper) could be added (e.g., in the "Conclusions" section). These recommendations could include the application of the methodological framework of this work to other areas around the globe. Discussions on the minimum observed data availability requirements for such an application would be also beneficial, to my view.*

We agree with the reviewer. Accordingly, the following sentence has been added to the conclusions of the paper:

"Our results suggest that validation at local scale of GCM simulations is an essential step to inform downscaling procedures and correction techniques, to make sure that model predictions are consistent with the local features of climate. However, extreme events like multiyear droughts are unfrequent and therefore validating their predicted statistics is particularly challenging.

Therefore, the identification of future drought risk, which one would expect to be increased under climate change, remains a challenge, especially if we consider that the reliability of bias correction depends on the availability of observed historical data. For some situations, classical engineering methods for critical event estimation under the assumption of stationarity, with appropriate integration with climate models to account for climate change, may still be the most precautionary option to make a synthesis of the available information."

*3) In line 233, it is written that "the ensemble mean performs better than any individual model for all indices". However, according to Figure 10, the ensemble mean performs better than the individual models for most, but not all, indices. Note that, for instance, it exhibits worse performance than GFDL-ESM4 in terms of the index referring to very heavy rainfall days (R20mm).*

We agree with the reviewer. We also agree with Editor's comment (see below) regarding the potential bias that may be induced in the extreme value by subgrid variability, as convective events may play a significant role in the dynamics of extremes. Thus, in the revised version of the paper, we have removed the analysis of the extremes (previous sections 3.3 and 5.1.3; we also removed previous Figure 10). The analysis of the extremes, in fact, is not relevant to drought analysis, which is the main focus of our paper.

*4) Figure 9 could be extended for providing information about the seasons December-January-February (DJF) and June -July-August (JJA) as well.*

We agree with the reviewer. We have extended the figures for seasons DJF and JJA (new Figures 9 and 10) and also added the corresponding description for the result in the main text.

*5) Some amendments could be applied to Figures 4, 5 and 7 for improving their readability. Specifically, for all the sub-figures belonging to each of these figures, the axes limits could be set the same. Moreover, a note could be added to the caption of Figure 8 for explaining that the thick lines represent the observed time series and the ensemble mean or, even better, the legend could be amended for providing this information. Lastly, the text size in Figure 6 could be increased.*

Thanks for the suggestion. The address on of the concerns of the Editor (see below our reply to Editor's comments), we have changed the part of the methodology and thus removed the previous Figures 4, 5, and 6. We have adjusted the axes for sub-figures of the previous Figure 7 (now Figure 6 in the revised manuscript) to be more readable. We have modified the legend of the previous Figure 8 and used the thick lines to represent the observed series and the ensemble mean, which can be found in the new Figures 7 and 8.

**Reply to Anonymous Reviewer #2**
We thank the reviewer for the thorough and helpful comments. They are very useful to improve the clarity of our manuscript. Here below we explain how the comments of the reviewer will be addressed.

*Page 1, line 16 and page 6, line 122: Check for the references so that they appear in chronological order.*

We have checked the references and made the required amendments.

*Page 14, Figure 8: The lines representing both, the observed and ensemble data, should be thicker in the legend to really appear as they have been plotted in the figure.*

Thanks for your suggestion. We have modified the legend of the previous Figure 8 and used the thick lines to represent the observed series and the ensemble mean, which can be found in the new Figures 7 and 8. (see also our reply to comment #5 by reviewer #1).

*Page 15, Figure 9: I suggest changing the symbol of the ensemble set (maybe a star or a square) to be easily identified. As it is now it is difficult to differentiate it from the ACCESS-CM2 set.*

We agree with the reviewer and therefore have changed the symbol of the ensemble set into a star to be easily identified (see new Figures 9 and 10).

*It would be interesting to know $R_{LT}$ values by maybe including them in Table 4.*

Thanks for your suggestion. $R_{LT}$ values represent the long-term mean value of annual rainfall for each model and observed data. We have included the value for observed data in revised manuscript in section 6.1.3 as follows:

"For the observation data, the long-term mean value of annual rainfall, which $R_{LT}$ of 705 (mm/year) is considered as the threshold to identify the observed multiyear drought events."

However, in previous Table 4, we want to show the statistics for multi year drought events to be more focused on drought. Therefore, we prefer to maintain the current format of the table.

**Reply to Anonymous Reviewer #3:**
We thank the reviewer for the thorough and helpful comments. They are very useful to improve the clarity of our manuscript. Here below we explain how the comments of the reviewer will be addressed.

*The manuscript is well written and the tables and figures are useful and adequately presented. My only comment refers to the fact that the use of historical time series to predict the future occurrence and characteristics of multiyear drought is not really included in the work, which should be more explicitly said when retrieving conclusions on the comparison of historical time series and future projections by GCMs.*

The comment of the reviewer is appropriate. To address it, we have changed the following text in the conclusions of the paper: "For some situations, classical engineering methods for critical event estimation under the assumption of stationarity may turn out to be more precautionary." as follows:

"For some situations, classical engineering methods for critical event estimation under the assumption of stationarity, with appropriate integration with climate models to account for climate change, may still be the most precautionary option to make a synthesis of the available information."

*Lines 149-150. Please, include some comment on the choice of these threshold values. Are they scaled in Figure 3?*

To address the comment of the reviewer the following sentence have been added at the end of line 150 of the original manuscript: "The threshold values have been identified with a trial and error procedure by verifying that relevant droughts observed in the past have been consistently identified." The threshold values have also been scaled in Figure 3.

*Lines 193-196. This paragraph starts by including all models in the same category, with poor capability to represent annual rainfall during the historical period, including their ensemble result, but the final sentence points out to the latter reproducing the mean of the observations. I suggest to redact this more clearly.*

We agree with the reviewer. We would like to point out that the comparison between observed and simulated statistics has been restructured to address concerns by other reviewers. Therefore the presentation of the analysis is now revised. The new version has been prepared by taking into account the above comment by the reviewer.

*Figures 6 and 7. Please, add "of annual rainfall" in the captions.*

Thanks for the suggestion, we fully agree with it and apologize for the lack of clarity. The previous figures 6 and 7 have been substituted by new figures 4 and 5, for which we now provide clearer captions.

*Figure 8. Please, increase the width of the lines in the legend for observations and ensemble to facilitate their identification in the graph.*

We agree with the reviewer. We have modified the legend of the previous Figure 8 and used the thick lines to represent the observed series and the ensemble mean, which can be found in the new Figures 7 and 8.

*Table 4. Are DD and DI mean values during the studied period? I suggest writing "Some statistics" in the caption, instead.*

We have modified the caption of previous Table 4 to the new Table 3 as follows:

"Table 3. Mean values over the considered period of drought frequency (DF), duration (DD), intensity (DI) and maximum deficit (MD) for multiyear meteorological droughts exhibited by observed data (1850-2014) and reproduced by models for the historical (1850-2014) and future (2015-2100) periods under the two considered scenarios before bias-correction."

We include a similar caption for our new Table 4.

*Line 245. I would drop the use of "significantly" here, since no significance test is really done, even if the values show this apparent difference. This also holds in other places in the text (e.g. line 267).*

We agree with the reviewer. Accordingly, we have dropped the word "significantly" in line 267 and replaced "significant" with "evident" in line 334. We have dropped "significant" in line 290. We have also replaced "significant" with "considerable" in line 281, "significantly" with "considerably" in line 335. The line number may be not consistent with the previous version due to some sentences have been deleted and also new results added into the context.

*Figure 11. The 30-yr moving average for the projections under the different future climate scenarios could also be added as in the historical observations.*

We agree with the reviewer that the previous Figure 11 is not easy to read. To avoid including additional lines that may distract the reader, we have removed the moving average line for historical data.

*Line 285. I would write " of SOME statistics", not so general as it is in the text.*

We agree with the reviewer and have amended the text accordingly.

*Lines 290-294. Related to my previous comment on Table 4, these sentences would then refer to mean values and, thus, these comments should clarify that less critical mean behaviour are produced by models, although extremes are not assessed. This might also affect the run theory application if alternating extremes take place, resulting in less drought events being identified in the future projections.*

We agree with the reviewer and therefore changed the text which can be found in the revised conclusion as follows:
"In fact, our focus is concentrated on the statistics of multiyear droughts. We found

that the multi-model ensemble can satisfactorily simulate the mean frequency of drought during the historical period. Conversely, the mean duration, intensity, and maximum deficit of multiyear drought are underestimated."

**Reply to Anonymous Reviewer #4:**
We thank the reviewer for the thorough and helpful comments. They are very useful to improve the clarity of our manuscript. Here below we explain how the comments of the reviewer will be addressed.

*Section 3.1 Authors describe the cumulative annual rainfall comparison. I would avoid to mention "hyetograph" since, for a moment, I was disoriented thinking that the single hyetograph within the years were included in the analysis. Maybe simply "time series" would be appropriate.*

We agree with the reviewer and have changed "hyetograph" with "time series".

*Section 3.2. While in Section 3.1 detailed information and equations are provided (Eq. 1-6), in this Section details are not provided as well. I would give more details on the Taylor diagram, on how to read it and explaining the three considered parameters.*

We have added a few more details to interpret the Taylor diagram in section 4.2 as follows:

"The angular coordinate represents R. The CRMSE is measured by the distance from the point of reference (observation). Finally, the radial distance from the origin represents the ratio of SD between model simulation and observation. For perfect model simulation, R and the SD ratio assume unit value and CRMSE is equal to 0." and have invited the reader to refer to Taylor (2001) for more information.

*Section 3.3. Authors compare eight extreme rainfall indexes using RMSE, however they limit the evaluation in comparing each model respect to the others not offering the single performance. I would suggest to add a plot to the Figure 10 that allows the reader to figure out the single model attitude to correctly simulate extreme values. A Relative Absolute error could be appropriate.*

We understand the concern of the reviewer. However, in view of the concern by other reviewers that that neglecting subgrid variability may be not justified for high rainfall events, which originate from convective processes, in the revised version of the paper, we have removed the analysis of the extremes (sections 3.3 and 5.1.3, also previous Figure 10). See also our reply to comment #3 of referee #1.

*Section 5, 6 , and 7. I am impressed by the results. Above all by Figure 4, 5, 6, and 7. Table 3 as well. Authors well commented the results, however in the conclusion they*

*could be clearer. Indeed the sentences present in lines 284, 286, and 302 seem in contradiction to the obtained results. From Figure 4 and 5, ensemble plots can not deserved any reliability, from Figure 6, the minimum MARE is more than 20%, finally looking Figure 7 and Table 3, moments and autocorrelation are always misunderstood. It is clear to me that, for Bologna time series, the GCM are not capable to reproduce rainfall and this confirmed in the drought analyses and related conclusions.*

We agree with the reviewer and therefore changed the text in the conclusions as follows:

"The present study refers to the region of Bologna, where the availability of a 209-year-long daily rainfall series allows us to make a unique assessment of GCM reliability and their predicted changes in rainfall and drought risk.
The results show that GCMs provide a satisfactory simulation of rainfall seasonality while statistics of rainfall series estimated for the long term historical period exhibit discrepancies among models and limited reliability in some cases. In particular, the correlation of annual rainfall looks underestimated, thereby implying a possible lack of fit in the simulation of long term cycles.
In fact, our focus is concentrated on the statistics of multiyear droughts. We found that the multi-model ensemble can satisfactorily simulate the mean frequency of drought during the historical period. Conversely, the mean duration, intensity, and maximum deficit of multiyear drought are underestimated.
Bias correction improves the simulation of the statistics of the monthly and annual series while it does not show consistent enhancements in capturing the correlation of annual rainfall and the distribution of seasonal rainfall.
The improvement by QDM to simulate drought characteristics is limited. Indeed, future projections by the multimodel ensemble of multiyear droughts depict a similar risk as in the past, even after bias correction and adopting the most critical emission scenario.
Our results suggest that validation at the local scale of GCM simulations is an essential step to inform downscaling procedures and correction techniques, to make sure that model predictions are consistent with the local features of climate. However, extreme events like multiyear droughts are unfrequent, and therefore validating their predicted statistics is particularly challenging.
Therefore, the identification of future drought risk, which one would expect to be increased under climate change, remains a challenge, especially if we consider that the reliability of bias correction depends on the availability of observed historical data. For some situations, classical engineering methods for critical event estimation under the assumption of stationarity, with appropriate integration with climate models to account for climate change, may still be the most precautionary option to make a synthesis of the available information. Therefore, rigorous use and comprehensive interpretation of the available information are needed to avoid mismanagement, by also taking into account that the impact of multiyear

meteorological droughts is likely to be exacerbated by further pressure on water resources due to increasing population and water demand."

**Reply to the Editor's comment**
We would like to thank the Editor for the careful assessment of our contribution. It is constructive and helpful to improve our presentation. The Editor raised two major concerns that are summarised here below in italic:

1) *"I suggest the authors to re-consider the sections of their work that focus on comparing different statistics between the historical observations and GCM simulations, rather than comparing concurrent time series."*
2) *"They should also carefully consider the potential biases introduced by the area to point estimation issue."*

Regarding the first issue, we fully agree with the Editor that GCMs should be evaluated by assessing their capability of reproducing the statistics of observed data, including the progress of statistics in time. Particularly in our case, we believe it is important to assess whether the evolution along time of rainfall statistics in Bologna is well reproduced by GCMs. In fact, the only comparison of the probability distribution of annual data over the full observation period does not provide enough information on the capability of models to predict how climate will change in the future. For instance, the probability distribution would not change if the sequence of rainfall is shuffled therefore eliminating change and persistence. To make a comprehensive assessment of the capability of models to reproduce change, it is also necessary to present a comparison of statistics for common subperiods.
However, we recognize that the annual subperiod may be too short for a meaningful assessment of statistics (note: annual rainfall is a statistic computed on the observed and simulated daily observations), and therefore we recognize the potential weakness of our approach in this respect.
To resolve such weakness, in the revised version of the paper we make the comparison between observed and simulated statistics of monthly data by referring to 33-year long time windows, instead of the annual window. We also revised the wording through the paper to better emphasise that we are comparing statistics for common subperiods and not observations.
Furthermore, we wanted to comply with the following suggestions by the Editor:

*"Comparative indices based on measuring simultaneous historical observations with simulations as the Kling-Gupta Efficiency criteria, would not be appropriate as well as the MARE and NSE indices"*

and

*you might want to consider a good number of statistics covering different rainfall*

*properties other than cumulative rainfall amounts.*

Therefore, in the revised version of the paper we use the the "Combined Probability-Probability" (CPP) plot (Koutsoyiannis and Montanari, 2022) to assess the performance of each of the considered CMIP6 GCMs in reproducing the statistics of monthly rainfall data within each 33-year long windows of the historical period (1850-2014). It is based on the comparison of the marginal probability distributions of observed and simulated data. CPP plot compares the probability distributions of observed and GCM simulated monthly rainfall, respectively, during the historical period. The comparison refers to 5 subsequent 33-year long time windows during 1850-2014 to assess the capability of GCMs to reproduce changes along the time of climate statistics. (See the detailed description in the new section 4.1).

Regarding the second issue which refers to potential bias between GCM simulations and observations, we agree with the Editor that subgrid spatial variability may be underestimated by interpolating grid rainfall, as convective rainfall may be not well reproduced at the grid scale. However, we analyse monthly, seasonal and annual rainfall, whose variability in space for the Bologna region and the considered grid size can be assumed to be negligible. Support to the above assumption is provided by the annual climatic reports by the Regional Agency of Environmental Protection, which are presented at https://www.arpae.it/it/temi-ambientali/meteo/report-meteo/rapporti-annuali for the past 5 years. Each of these reports presents maps of the spatial distribution of each year's cumulative rainfall over the region. Such maps show that the variability is essentially governed by ground elevation, whose variabioity is limited in the region around Bologna which is essentially flat. Therefore we believe that our procedure does not introduce a systematic bias. The study presented by Antolini et al. (2016) also confirms the low variability in space for spatial and long-term seasonal rainfall as well.

However, to provide better support to the reliability of our analysis we decided to use bi-linear interpolation of rainfall from the 4 grid points around the location of Bologna to estimate point rainfall. We selected bi-linear interpolation after trying different spatial interpolation methods such as weighted inverse distance and nearest-neighbour interpolation and checking that the results did not change much. Bilinear interpolation is also commonly used to alleviate the scale problem of a mismatch between the coarse grid and station point. (Bracegirdle and Marshall, 2012; Zhang et al., 2022).

Furthermore, to compensate for potential bias we expanded our analysis by including an assessment of bias-corrected GCM predictions (Section 3 of revised manuscript). We use quantile delta mapping (QDM) to correct bias with respect to observed statistics.

We quote from the new Section 3 of the revised paper:

"Simulations by GCM are provided at the grid scale. To compare them with observed data, one should take into account the potential bias that may be introduced by subgrid variability. For the considered time scale subgrid variability is expected to be limited in the region of Bologna. In fact, we focus on monthly and annual rainfall data, which exhibit low spatial variability in the region (see the annual reports of the Regional Agency for Environmental Protection at https://www.arpae.it; see also Antolini et al. (2016) for an analysis of subgrid variability in the considered spatial domain).
To compensate for potential bias, we applied bilinear spatial interpolation to estimate the model prediction for Bologna depending on the four nearest GCM grid points. Moreover, we applied quantile delta mapping (QDM) to correct bias with respect to observed statistics." (detailed description of QDM can be also found in section 3)".

Furthermore, by recognizing that subgrid variability may play a relevant role for high rainfall events, which originated from convective processes, in the revised version of the paper we removed the analysis of the extremes (sections 3.3 and 5.1.3), which we also recognize is not very relevant for drought risk assessment.
To summarise, we made the following changes to the revised manuscript to resolve the concerns of the Editor:
1) Substituted the comparison of annual rainfall statistics with 33-years monthly rainfall statistics by using CPP plot;
2) Provided better evidence for the limited subgrid variability of long term rainfall in the region, a more detailed description of the interpolation method, and included an additional analysis by applying QDM to correct potential bias in the simulations;
3) Removed the analysis of rainfall extremes;
4) Revised the wording in the paper according to the above changes, by particularly emphasizing the value and interest of comparing rainfall statistics computed over the whole observation period and common subperiods.
In particular, we believe that items 2 and 3 above provide support to the comparison of the statistics of GCM simulations with those of one of the longest rainfall series today available at the global level.

References

Antolini, G., Auteri, L., Pavan, V., Tomei, F., Tomozeiu, R., & Marletto, V. (2016). A daily high-resolution gridded climatic data set for Emilia-Romagna, Italy, during 1961–2010. International Journal of Climatology, 36(4), 1970-1986.
Koutsoyiannis, D., & Montanari, A. (2022). Bluecat: A local uncertainty estimator for deterministic simulations and predictions. Water Resources Research, 58(1), e2021WR031215.
Bracegirdle, T. J. and Marshall, G. J.: The Reliability of Antarctic Tropospheric Pressure and Temperature in the Latest Global Reanalyses, Journal of Climate, 25, 7138–7146,

https://doi.org/10.1175/JCLI-D-11-00685.1, 2012.

Zhang, H., Zhang, F., Zhang, G., and Yan, W.: Why Do CMIP6 Models Fail to Simulate Snow Depth in Terms of Temporal Change and High Mountain Snow of China Skillfully?, Geophysical Research Letters, 49, e2022GL098888, https://doi.org/10.1029/2022GL098888, 2022.

---

## Editor Decision (ED1)

**Dear Authors,**
**Thank you very much for your responses to my comments. According to your responses, you are planning to do the following:**

1) *Substitute the comparison of annual rainfall statistics with 10-years rainfall statistics;*

**This might lead to an improvement of your paper presentation and correct initial flaws, but you might want to consider a good number of statistics covering different rainfall properties other than cumulative rainfall amounts.**

2) *Provide a discussion of subgrid variability and an expanded description and discussion of our assumption in this respect and the interpolation method that has been used.*

**This might represent an improvement of your results since spatial variability would be accounted for in your analysis. However, it will be difficult to support reasonable conclusions about this issue, since you are relying on the analysis of a single location.**

3) *Remove the analysis of rainfall extremes;*

***Agreed***

4) *Revise the wording in the paper according to the above changes, by in particularly emphasizing the value and interest of comparing rainfall statistics computed over the whole observation period and common subperiods.*

***Agreed***

***In view of the important number of changes in the conceptualization of the research and the need to add a good number of new analysis and results, I would reject the paper and encourage the authors to re-submit again after considerations of all suggestions and improvements.***

---

## Author Response (AR2)

**Historical rainfall data in Northern Italy predict larger meteorological drought hazard than climate projections**

Rui Guo and Alberto Montanari

Department of Civil, Chemical, Environmental and Materials Engineering (DICAM), University of Bologna, Bologna, Italy

Correspondence: Rui Guo (rui.guo2@unibo.it)

We are grateful to the Editor for managing the review process of our paper and the positive feedback for our previous revision. We are also grateful to the additional reviewer for the positive comments and further suggestions which helped us to improve the manuscript.

Within this rebuttal document we describe how the remarks by the Reviewer were addressed. Quotes from the reviewer are reported in italic. We include here below selected quotes from the revised manuscript to better explain how the related concerns were addressed. Quotes from the revised manuscript are highlighted in blue.

**Reply to Anonymous Reviewer #4**

We thank the additional reviewer for the positive feedback of our previous revision and the helpful review of our manuscript. Here below we explain how the comments of the reviewer had been addressed. Comments are quoted in italic.

*- Eq. 1. I am not sure if it correct, Should mh(t) be the denominator?*

Thanks for your comment. We checked the Eq. 1 and it is correct. In this equation, it is true that the denominator is the value in historical period. However, it cannot be directly represented by *mh(t)* due to that *t* represents future period in this equation. This equation uses the empirical frequency of not exceedance in future period qf (t) to represent the same quantiles in historical period and then uses the distribution of empirical frequencies in historical period to calculate the correspongidng value. We deleted *mf(t)* and *mh(t)* in Line 101 to avoid misleading.

*- Line 260. In my opinion, it is not just an underestimation. The GCM time series are simply random. I would not be surprised if the ACF_lag1 coefficients would be not significant for all 13 scenarios. This is crucial for drought analysis, of course, since in the observed time series the rainfall amount collected in one year is linked to the amount collected in the following year. Since the coefficient is only 0.22 I expect that the observed time series show data "grouped" in pairs or triplets while the GCM ones are decorrelated.*

*Maybe the authors could better specify this aspect.*

We agree with the reviewer. To better specify this aspect, we modified the line 260 as follows:

*"It is interesting to note that the correlation of observed data is slightly higher than all the models, therefore highlighting possible model weakness in simulating temporal correlation. Since the coefficient of the observation series is 0.226, the observed values may be correlated in pairs or triplets while most of the GCMs' series are uncorrelated, thereby implying a possible inadequacy of bi-annual or triennial fitness."*

*- Figure 8. I would use the same y-axis range of Figure 7, that is 0-180 mm*

We agree with the reviewer. The y-axis of Figure 8 has been changed to the same range of Figure 7.

*- Line 396. The same previous comments concerning the underestimation and in addition, I would not say "long term cycle" but "short term cycle (bi-annual, triennial)".*

We agree with the reviewer. The sentence in Line 396 has been change into:

*"In particular, the GCMs show weakness in capturing the correlation of annual rainfall, thereby implying a possible lack of fit in the simulation of cycles."*